# COVID-19: Variants, Immunity, and Therapeutics for Non-Hospitalized Patients

**DOI:** 10.3390/biomedicines11072055

**Published:** 2023-07-21

**Authors:** Cameron Y. S. Lee, Jon B. Suzuki

**Affiliations:** 1Private Practice in Oral, Maxillofacial and Reconstructive Surgery, Aiea, HI 96701, USA; 2Department of Periodontology and Oral Implantology, Kornberg School of Dentistry, Temple University, Philadelphia, PA 19140, USA; 3Department of Graduate Periodontics, University of Maryland, Baltimore, MD 20742, USA; 4Department of Graduate Prosthodontics, University of Washington, Seattle, WA 98195, USA; 5Department of Graduate Periodontics, Nova Southeastern University, Fort Lauderdale, FL 33314, USA; 6Department of Microbiology and Immunology, School of Medicine, Temple University, Philadelphia, PA 19140, USA

**Keywords:** COVID-19, variants, immunologic response, vaccines, therapeutics

## Abstract

The continuing transmission of coronavirus disease 2019 (COVID-19) remains a world-wide 21st-century public health emergency of concern. Severe acute respiratory syndrome coronavirus 2 (SARS-CoV-2) has caused greater than 600 million cases of COVID-19 and over 6 million deaths globally. COVID-19 continues to be a highly transmissible disease despite efforts by public health officials and healthcare providers to manage and control the disease. Variants identified in selected worldwide epicenters add to the complexity of vaccine efficacy, overage, and antibody titer maintenance and bioactivity. The identification of the SARS-CoV-2 variants is described with respect to evading protective efficacy of COVID-19 vaccines and breakthrough infections. Vaccines and other therapeutics have prevented millions of SARS-CoV-2 infections and thousands of deaths in the United States. We explore aspects of the immune response in a condensed discussion to understand B and T cell lymphocyte regulatory mechanisms and antibody effectiveness and senescence. Finally, COVID-19 therapies including Paxlovid, Remdisivir, Molnupiravir and convalescent plasma in non-hospitalized patients are presented with limitations for identification, collection, and distribution to infected patients.

## 1. Introduction

Coronavirus disease (COVID-19), caused by severe acute respiratory syndrome coronavirus-2 (SARS-CoV-2), was first diagnosed in December 2019, and has progressed to a global pandemic [1,2]. The continuing transmission of coronavirus disease 2019 (COVID-19) remains a worldwide 21st century public health emergency of concern. Older individuals, smokers, and those with comorbidity risk factors such as cardiovascular disease, chronic respiratory disease, cerebrovascular disease, chronic kidney disease, chronic liver disease, cancer, diabetes mellitus, and obesity are less able to mount antibody responses and are, therefore, at increased risk for adverse outcomes, inpatient hospitalization, and death [3,4,5,6,7].

In a meta-analysis by Wu and McGoogan [8], patients with comorbidity risk factors were twice as likely to progress to severe COVID-19 and die compared to patients without those comorbidities. Older adults with preexisting comorbidities were also at greater risk of mortality [5,6,7,8].

Severe acute respiratory syndrome coronavirus 2 (SARS-CoV-2) has caused greater than 600 million cases of COVID-19 and over 6 million deaths globally [1]. COVID-19 vaccines have prevented millions of SARS-CoV-2 infections and thousands of deaths in the United States [2].

Variants of concern, immunity, vaccine effectiveness and other therapeutics are the focus of this paper. Currently, the B.1.1.529 (omicron) variant of severe acute respiratory syndrome coronavirus 2 (SARS-CoV-2) has mutated into several subvariants with the capability for increased viral transmission and immune escape ability [1]. Omicron subvariant BA.5 is the dominant global virus and has demonstrated a greater ability for immune escape compared to previous subvariants of omicron [9]. In a study by Iketani et al. [10] that evaluated neutralizing antibody titers against five coronavirus strains (WA1/2020, omicron subvariants BA.1, BA.2, BA.4, BA.5 and BA.4.6) their data revealed that BA.4.6 omicron subvariant easily escaped neutralizing antibodies induced by prior infection or vaccination. These data suggest the continued mutation of SAR-CoV-2 and increasing prevalence in populations where BA.5 is dominant, including the United States.

## 2. COVID-19 Public Health Variants of Concern

Since COVID-19 was discovered in Wuhan, China, [11] the coronavirus has mutated into many new variants with greater transmissibility and the ability for immune escape [12,13]. By the end of 2020, Alpha (B.1.1.7), beta (B.1.351), and gamma (P.1) variants were discovered in the United Kingdom, South Africa, and Brazil, respectively. In 2021, the dominant global variant of concern was B.1.617.2 (delta) which was discovered in the summer of 2021 in India (Table 1). Figure 1 details SARS-CoV-2 variants with a focus on spike proteins.

This was soon replaced by the omicron variant (B.1.1.529) by the end of 2021 which emerged in Africa and is now a dominant variant of concern worldwide [15]. The dominance of the omicron variant is due to its ability to mutate over 50 times in both the spike protein and the receptor binding domain, which is the main target of neutralizing antibodies. Such escape ability contributes to its ability for neutralizing antibody escape despite prior infection, vaccination, or hybrid immunity [16,17,18]. To control the continued genetic evolution of SARS-CoV-2, the US Food and Drug Administration recently approved the use of bivalent booster vaccines to reduce the incidence of severe disease, hospitalizations, and death [19,20]. The following subvariants have been discovered and include BA.1, BA.1.1, BA.2, BA.2.12.1, BA.4, and BA.5, BF7, BQ1.1.and XBB [10,15]. The variants that have propagated the past waves are all members of the SARS-CoV-2 family that have developed subvariants that continue to increase the waves of infection.

Of the different subvariants of concern, healthcare systems around the globe should be vigilant for XBB as it has demonstrated the ability to be extremely transmissible and resistant to neutralizing antibodies greater than BA.5 [21]. Currently, XBB has been detected in 35 countries and makes up 54% of COVID-19 cases in Singapore, and 18.4% in the United States [22]. This subvariant has demonstrated increased resistance to antiviral humoral immunity from breakthrough infections by a unique evolutionary pathway [23].

Using phylogenetic analyses, Tamura, and colleagues [23] discovered that the XBB virus is a recombinant virus as two distinct genomes have merged together rather than occurring by convergent evolution (random mutation). XBB emerged from two subvariants of BA.2 (BJ.1 and BM.1.1.1). Such shared genetic material occurs on the receptor binding domain (RBD) of the viral spike protein. However, increased antiviral immunity could also be due to genetic mutations beyond the RBD. Mutations have also been observed on the N-terminal domain (NTD).

Because of omicron subvariants’ immune evasive strains, the FDA advisory committee authorized the use of bivalent vaccines. Bivalent booster vaccination has been shown to increase neutralizing antibodies to the omicron variant compared to monovalent boosters [18,19,24,25]. Altarawneh and colleagues [25] reported that a third booster resulted in an approximate 60% reduced risk of infection if boosting occurred within 45 days from the previous immunization. Other studies reported that after a third mRNA immunization, decreased clinical protection and waning were observed at 4 months [25,26]. More striking, after a fourth mRNA immunization waning was reported after just 4 weeks. However, protection against the severe form of COVID-19 was clinically observed that prevented admission to the intensive care units of hospitals and deaths [21,22]. The protection conferred by hybrid immunity to SARS-CoV-2 provided the best long-term protection compared to either natural immunity or vaccination [16,24]. This finding suggests the benefits of vaccination against COVID-19-related hospitalizations and death.

## 3. The Innate and Adaptive Immunological Response to COVID-19

The immune system is the human body’s defense against pathogens, such as bacteria, viruses, fungi, parasites, toxins, and cancer cells and consists of innate and adaptive responses [27]. Vaccines are designed to condition the immune system to protect the human body from infections due to immunologic memory. As immunology is a broad subject, it is beyond the scope of this paper to provide a comprehensive review. Instead, it is the intent of this paper to provide public health professionals and other healthcare providers with a basic introduction to the function of the immune system in disease and SARS-CoV-2.

Innate immunity is the first line of resistance (non-specific) to invading microbial pathogens by recruiting immune cells to the site of inflammation and infection, such as type I interferons, cytokines, neutrophils, monocytes, macrophages, and natural killer T cells [28].

Important cytokines recruited to the site of the invading pathogen include tumor necrosis factor (TNF), interleukin 1 (IL-1) and interleukin-6 (IL-6). Innate antiviral immunity is activated within hours of the invading pathogen and does not have the ability to recognize the same pathogen in future encounters [29]. It will also activate the adaptive immune response with antigen-presenting cells.

The adaptive immune response is antigen-specific and has the capability to recognize the same antigen in future encounters [30]. Adaptive immunity consists of two mechanisms- humoral and cellular immunity. With humoral immunity (antibody-mediated immunity), antibodies attach to the spike protein of the coronavirus to eliminate the virus from the human body. With the invasion of SARS-CoV-2, cellular immunity activates B cells and antigen-specific CD8+ T cells and CD4+ T cells to eliminate infected cells and block viral replication [31]. Humoral and cellular immune mechanisms are activated with SARS-CoV-2 infection. Both mechanisms attempt to prevent severe SARS-CoV-2 infection that results in hospitalization and death [32,33]. Research has revealed that approximately 95% of individuals maintain immune memory for about eight months due to natural infection from antigen-specific antibodies, memory B-cells and T-cells.

## 4. COVID-19 Vaccine Effectiveness

Vaccination is the most important way to protect individuals from COVID-19-associated hospitalizations and death [1]. Transmission of SAR-CoV-2 is due to replication and shedding of the virus in the upper respiratory tract. Therefore, vaccines must direct their mechanism of action to control replication and transmission of the virus. Globally, greater than 300 vaccines have been developed in response to COVID-19 [1]. However, only 10 COVID-19 vaccines have been approved by the WHO to mitigate viral disease caused by the Wuhan ancestral (D614G) strain which was soon replaced by other variants. These vaccines represent four different vaccine platforms: messenger RNA (mRNA); adenovirus vector-based; inactivated virus and adjuvanted protein vaccines. In the United States, the 2-shot four vaccines approved for use are the following: mRNA vaccines BNT162b2 and mRNA-1273; adenovirus vector-based vaccine Ad26.CoV2. S and adjuvanted protein vaccine NVX-CoV2373. Prior to the emergence of the omicron variant, in Phase 3 clinical trials, vaccines demonstrated 94–95% clinical efficacy (Table 2) against COVID-19 infection except for the 1-shot Ad26.COV2. S (72% efficacy) [34,35,36,37].

In the United States, mRNA vaccines (Moderna’s Spikevax mRNA-1273 and Pfizer-BioNTech; s BNT162b2) have primarily been used against the coronavirus with remarkable success [1]. In the United States, data from the Centers for Disease Control and Prevention (CDC) demonstrated that vaccination protected individuals during the delta variant surge during the winter months of 2020 and 2021 and spring of 2022 [37]. However, it was also observed that breakthrough infections were greater with the Ad26.COV2. S vaccine compared to the mRNA vaccines (Figure 2). The Food and Drug Administration and CDC have prohibited the use of Ad26.CoV2. S in the United States because of the number of adverse side effects observed. Vaccine-induced immune thrombotic thrombocytopenia (VITT) developed in 54 individuals and nine died [38,39]. The vaccine complication represents an incidence of three cases per one million vaccinated persons. VITT has also been reported in the United States in three patients (one died) who were vaccinated with the mRNA1272 vaccine [40,41]. Despite this adverse side effect, adenovirus vector-based vaccines are used in developing countries because of the lower cost and not requiring maintaining the vaccine at subfreezing temperatures.

Individuals in the US have immunity against SARS-CoV-2 from prior infection, vaccination, or hybrid immunity (combination of both) [42,43]. However, the SARS-CoV-2 Omicron variant (including BA.5) detected in November 2021 rapidly became the dominant circulating variant worldwide due to high transmissibility and immune evasion capability compared to the Delta variant. To better understand vaccine effectiveness data, Lin et al. [44] conducted a cohort study of 10.6 million people in the state of North Carolina who were vaccinated, and infected with the coronavirus, and compared them with unvaccinated individuals. In their study, 67% of the population were vaccinated and 2771.364 infections with the virus were reported. They also reported that hospital admissions were 6.3% and a mortality rate of 1.0%.

Based on the Lin et al. [44] study, several important conclusions may be made. With the low number of hospital admissions and deaths from COVID-19, vaccination is highly effective in preventing severe disease. Vaccination with boosters did not protect the public from milder infections. Further, prior infection with the coronavirus was associated with reduced risk of virus infection. For individuals with prior infection who also received the vaccine and booster, additional protection was observed from breakthrough infections. However, waning with booster vaccines occurred after 4 to 6 months.

## 5. Production of Novel COVID-19 Vaccines

Vaccination is the hallmark strategy to mitigate the COVID-19 pandemic through neutralizing antibody activity and plasma cell, B cell immunity, and T cell killer cells and immunological memory [45,46,47]. To bring the pandemic under control, many different new vaccines must be developed as each vaccine has different advantages and disadvantages against COVID-19. Each country with its different populations, public health care environments and age groups will be the beneficiary of different vaccine products that are developed on different platforms.

However, for at-risk populations such as the elderly and immunocompromised, the need for multiple vaccinations and boosters every 4–6 months may represent a public health challenge. Further, there is also the possibility that a new variant will emerge that will be entirely resistant to the current vaccines in preventing severe disease. The strategy to develop novel vaccines should be to produce a universal vaccine with long-lasting immunity that can be stored and transported at room temperature [45]. Such a novel COVID-19 vaccine would be similar to the influenza vaccine, where 75% of the population is expected to be protected from the influenza virus yearly and protects all age groups. Further, the novel vaccine should be a 1-dose vaccine per year based on an annual review of the current circulation of SARS-CoV-2 variants.

## 6. Current Therapies for SARS-CoV-2 Infections in Non-Hospitalized Patients

Clinical decisions for therapeutic interventions of SARS-CoV-2 are dynamic and likely to change with ongoing published research and clinical empirical observations. For this review on the pathophysiology of COVID-19 infections, two basic therapeutic strategies are condensed (see Figure 3):

The management of symptoms should be initiated for all non-hospitalized adults with mild to moderate COVID-19 [48]. For adults at high risk for progression to severe disease, several antiviral therapeutic options are available to decrease the risk of hospitalization or death. The information below for the management of COVID-19 is described below. The goal of management for non-hospitalized patients is to prevent progression to severe disease, hospitalization, and death. In selecting the best pharmacologic treatment available, several factors should be considered. Such factors are the following: clinical efficacy, availability of the medication, route of drug administration (parenteral or oral), possible drug-drug interactions, the onset of symptoms, pregnancy status, and in-vitro activities of medications against the circulating SARS-CoV-2 variants and subvariants.

Clinical trials of the treatment options at the time of this writing were conducted in unvaccinated individuals. Therefore, the vaccine efficacy in patients who have been vaccinated is unclear, and randomized clinical trials suggest a benefit [49,50,51,52,53].

In addition to vaccines, antivirals and monoclonal antibodies have been tried as treatments for SARS-CoV-2 infection but have proven challenging in reducing the risk of progression of severe disease and death [54,55,56]. Therefore, antiviral medications should be considered for COVID-19-positive patients who are at substantial risk (immunocompromised, over age 65 years, immunosuppressive medications) for severe COVID disease [48].

Pfizer’s Paxlovid, (Ritonavir/Nirmatrelvir) is an oral antiviral drug that obtained Food and Drug Administration Emergency Use Authorization (FDA, EUA) in December 2021 for patients with mild to moderate COVID-19, 12 years of age and older, who are at risk for progression to severe COVID-19 or mortality [57]. Paxlovid therapy consists of three doses with two doses of nirmatrelvir and one dose of ritonavir. Alone, neither nirmatrelvir nor ritonavir have therapeutic activity but are required to be sequentially administered for therapeutic value. Nirmatrelvir is a protease inhibitor that targets the 3-chymotrypsin-like cysteine protease enzyme of SARS-CoV-2 which is crucial in the viral replication cycle [58]. Ritonavir is an inhibitor of cytochrome P450 3A4 which decreases nirmatrelvir metabolism and increases blood serum levels.

In a phase 2–3 double-blind, randomized, controlled trial, Hammond and colleagues showed that administration of the two medications every 12 h for five days within three days of symptoms resulted in an 89.1% relative risk reduction in hospitalization and death [59]. Treatment with nirmatrelvir plus ritonavir also decreased the SARS-CoV-2 viral load by a factor of 10 by the fifth day of treatment compared to patients receiving the placebo. In a study by Najjar-Debbiny et al. [51] treatment with Paxlovid reduced the risk of progression to severe COVID-19 or death in both vaccinated and unvaccinated people.

However, there have been reports of recurrence of clinical symptoms after the completion of treatment with nirmatrelvir and ritonavir [60,61]. Although the frequency of recurrence is unknown, viral load rebound may be a characteristic of SARS-CoV-2 infections [62]. Because viral shedding can occur during the time of relapse, this reinforces the importance of COVID-19 testing and isolation for individuals who experience recurrent symptoms after treatment with nirmatrelvir and ritonavir.

Currently, Remdesivir (Veklury, Gilead Sciences) is the only FDA-approved antiviral drug for the treatment of COVID-19 [57]. Originally developed to treat Ebola virus disease, Remdesivir is a direct-acting nucleotide pro-drug inhibitor of the SARS-CoV-2 RNA-dependent RNA polymerase terminating viral replication in human airway epithelial cells [63,64]. In a phase 3 clinical trial, Remdesivir decreased the recovery time in patients hospitalized with COVID-19 in both a 10-day and 5-day antiviral course [65,66].

Gottlieb and colleagues conducted a randomized, double-blind, placebo-controlled trial involving 562 patients who received a 3-day course of Remdesivir had an 87% risk reduction of hospitalization or death from any cause at day 28 [67]. This clinical trial agrees with the Adaptive COVID-19 Treatment Trial (ACTT-1) that demonstrated that patients with moderate to severe COVID-19 treated with Remdesivir had a shorter recovery time and a risk reduction to progressing to severed COVID-19 compared to patients who received the placebo [65]. Further, when Remdesivir treatment is initiated early during COVID-19, data from the ACTT-1 trial showed that Remdesivir may prevent disease progression.

Remdesivir should be started within 7 days of symptom onset and administered for three days. Although Remdesivir is administered intravenously, it can be transported and stored at room temperature [63]. For patients throughout the world that do not have access to vaccines, Remdesivir may be an important therapy in the management of COVID-19 [67].

Merk’s Molnupiravir (Lagevrio) received FDA (USA) emergency use authorization in October 2021 for outpatient treatment of patients with mild to moderate COVID-19 who are 5 days of onset of symptoms, or at high risk for hospital admissions and death [68]. Originally developed against RNA viruses, in phase 2 clinical trials Molnupiravir was able to accelerate clearance of SAR-CoV-2. Molnupiravir 800 mg is taken orally for five days in non-hospitalized patients. In clinical trials, Molnupiravir demonstrated antiviral activity against SARS-CoV-2 by inducing viral mutations and mutagenesis blocking viral replication [69,70]. However, the orally administered antiviral medication achieved only a 30% reduction in COVID-19-related hospitalizations and death over a 29-day randomization period [71,72]. It is authorized for use only in patients who do not have access to FDA-authorized COVID-19 treatment aged 18 years and older as it may affect bone and cartilage growth. It is contraindicated in pregnant females due to abnormal fetal growth effects.

Some patients in selected countries or regions may not have access to vaccines or anti-viral therapies, i.e., Molnupiravir, Remedisivir, Paxlovid. Therefore, other strategies including the use of N95 masks, social distancing, air purifiers, and anti-viral mouth rinses may be useful in COVID-19 disease prevention and reduction.

Neutralizing monoclonal antibodies have demonstrated risk reductions of 70 to 85% against mild to moderate COVID-19 in outpatient clinical trials [73,74,75]. However, due to mutations on the spike protein, partial or complete resistance to monoclonal antibodies has been observed [76,77] resulting in greater virulence and viral transmission [78,79,80]. Therefore, monoclonal antibodies are not currently authorized for use in the United States because it is anticipated that the Omicron subvariants will not be susceptible to this form of therapy [81].

## 7. Convalescent Plasma

Passive immunity (passive antibody transfer) with convalescent plasma has been used for over 100 years in the treatment of infectious diseases until an immune response is activated in the infected patient [82,83,84]. During the 1918–1920 Spanish influenza A (H1N1) pandemic, convalescent plasma was extensively used because it was believed that plasma from donors who recovered from the influenza virus contained antibodies that may terminate viral replication and death [84,85]. This was confirmed in a meta-analysis by Luke and colleagues involving 1703 patients [84].

Before the development of antiviral and neutralizing antibody treatment during the start of the COVID-19 pandemic, the passive transfer of antibodies from the plasma of recovered patients who were infected with the coronavirus was one of the first treatments used to reduce the number of intensive care unit admissions, and ultimately death [86,87]. This is because many treatments such as anticoagulant, antiviral and anti-inflammatory medications were inconsistent producing controversial results [88]. Further, although neutralizing anti-spike monoclonal antibody treatment has been widely used in managing COVID-19 infections, the evolution of SARS-CoV-2 variants has resulted in greater virulence, viral transmission, and resistance [89].

Early during the COVID-19 pandemic, several studies reported on the use of COVID-19 convalescent plasma. However, the results were unclear as to the benefit of passive antibody transfer [90,91]. There is now more information on the role of high titers of neutralizing antibodies in decreasing the incidence of severe disease progression, hospitalization and death if administered within 72 h since the onset of symptoms [92,93]. Further, COVID-19 convalescent plasma maintains its clinical efficacy over time with new SARS-CoV-2 variants. Therefore, there is much interest in the clinical application of COVID-19 convalescent plasma, especially for patients who are immunocompromised and not able to mount a sufficient antibody response against the coronavirus and clear the virus.

However, accessibility to post-COVID-19 patients, selection of donors, lab preparation, and distribution continue to pose challenges for the use of convalescent plasma.

In addition, several scientific organizations (i.e., CDC/IDSA, AABB) have recommended the use of COVID-19 convalescent plasma in immunocompromised patients, especially after reports of monoclonal antibody-resistant SARS-CoV-2 variants [94,95]. In a retrospective study by Joyner and colleagues [96], they determined that the hyperimmune immunoglobulin and IgG antibody levels in convalescent plasma were associated with a lower risk of death in hospitalized patients compared to plasma with lower antibody levels. In a systematic review and meta-analysis by Senefeld et al. [93] they concluded that COVID-19 convalescent plasma was associated with a mortality benefit in immunocompromised hospitalized patients. These studies agree on the decreased mortality benefit with data from observational studies and randomized trials of high-titer antibody COVID-19 convalescent plasma [97,98,99]. While most studies on COVID-19 convalescent plasma were from unvaccinated donors, Vax-Plasma is now available for clinical use from regular donors. Vax-Plasma has been shown to retain higher neutralizing antibody titers and efficacy against the SARS-CoV-2 variants [100].

## 8. Virus Mediated Cellular Senescence as a Potential Therapeutic Target

The coronavirus disease (COVID-19) caused by SARS-CoV-2 has been responsible for over 6 million deaths globally [1]. Of the different risk factors for the development of COVID-19, advanced age was the most significant risk factor as the elderly population was disproportionately affected [101,102]. Cellular senescence is a primary determinant in the elderly developing COVID-19 where cells lose their ability to proliferate [103]. In a study by Fulop and colleagues [104] there is a direct correlation between aging and increased viral infection, such as hepatitis B virus (HBV), cytomegalovirus (CMV), human immunodeficiency virus (HIV), and human papillomavirus (HPV). This is due to viral infection causing an inflammatory response that can result in cellular senescence and has also been detected in patients infected with SARS-CoV-2 infection [105,106].

COVID-19 has also been observed to stimulate the release of chemokines and cytokines that can exacerbate immunosenescence via cytokine signaling [107]. A study by Chen et al. [108] reported that older patients infected with the coronavirus have a greater degree of senescent CD4^+^ and CD8^+^ T cells predisposing individuals to infection and poor response to vaccination. Treatment directed at senescence mechanisms such as cytokine storm and aging of T cells may be a strategy to reduce morbidity and improve vaccine effectiveness [109,110].

Further investigations are needed to understand viral mechanisms able to negatively impact on B and T cell senescence. Overcoming immune senescence in both healthy and immunocompromised and aging patients will certainly extend to T and B cell antibody production (currently between 4–6 months post-vaccine).

## 9. Vaccine Hesitancy

Vaccines have proven to be an effective treatment in preventing outbreaks of infectious diseases, including COVID-19 [111,112]. However, there are misperceptions that there is an elevated risk of infectious disease transmission because of the limited trust in vaccine science and research, and therefore, a low benefit of protection [113,114]. Such misinformation leads to vaccine hesitancy which will decrease the confidence that vaccines will protect the public. For COVID-19 vaccination campaigns to make progress in moving forward, healthcare personnel and community leaders are a reliable source of public trust about vaccine safety regarding the benefits of vaccination to society [115,116].

Side effects include pain, redness, and swelling around the site of infection (generally upper deltoid muscle area of non-dominant arm). More generalized adverse effects include tiredness, headache, muscle pain, chills, fever, and nausea.

In addition, the use of social media, and health information websites with accurate information regarding vaccine safety and efficacy could improve public acceptance for COVID-19 vaccination programs.

## 10. Conclusions

As the pandemic continues, oral and parenteral novel vaccines, antivirals and other therapeutics must be developed and have the ability for longer duration of plasma cell and B cell expression of antibodies. Strategies extending immunologic antibody titers against SARS-Co-V-2 would provide greater protection against viral transmission reducing hospitalization, morbidity, and mortality. The need for novel therapies combined with the substantivity of COVID-19 vaccines must be pursued to reduce the risk of hospitalizations and death caused by new and suddenly emerging coronavirus variations. In addition, such novel treatments must be able to be modified due to the threat of the emergence of new coronavirus variants to decrease the risks of hospitalizations and death. Such novel treatments would decrease the pressure on the healthcare system.

## Figures and Tables

**Figure 1 biomedicines-11-02055-f001:**
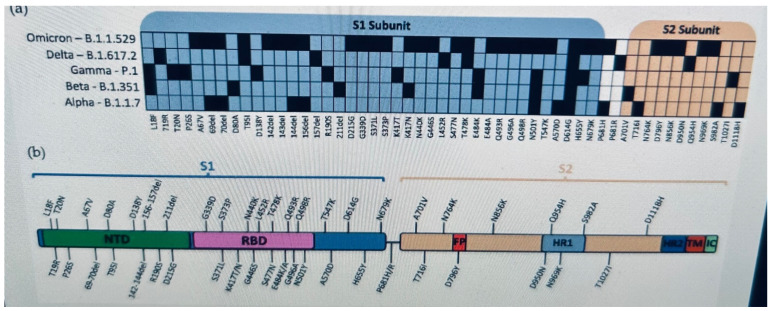
Mutations and Evolution of the SARS-CoV-2 Spike Protein, by Nicholas Magazine, Yingying Wu an Weishan. Adopted from *Viruses* 2022, *14*(3), 640. https://doi.org/10.3390/v14030640, Received: 30 January 2022/Revised: 12 March 2022/Accepted: 16 March 2022/Published: 19 March 2022 [14].

**Figure 2 biomedicines-11-02055-f002:**
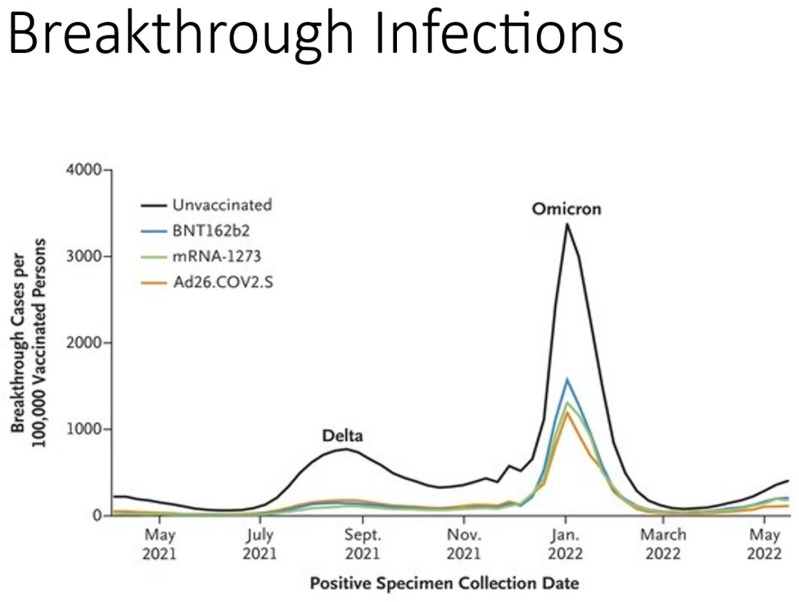
Protective Efficacy of COVID-19 vaccines in the United States. Breakthrough cases per 100,000 vaccinated cases who received the three vaccines from April 2021 to May 2022. Breakthrough infections to SARS-CoV-2 Delta and Omicron variants in patients vaccinated by current COVID-19 vaccines in USA. From the Centers for Disease Control and Prevention (https://covid.cdc.gov/covid-data-tracker/#rates-by-vaccines-status, accessed on 12 July 2023).

**Figure 3 biomedicines-11-02055-f003:**
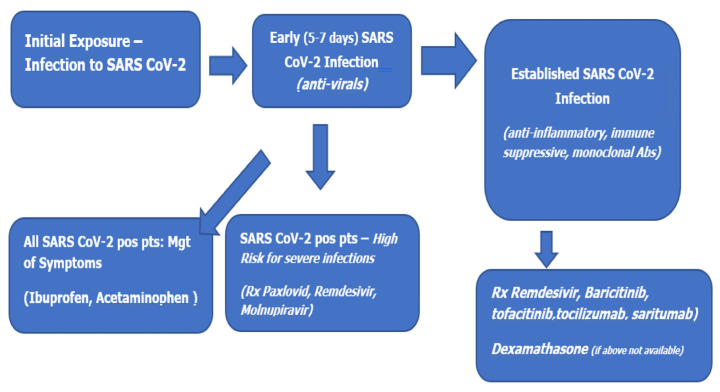
Treatment Recommendations for SARS-CoV-2 in non-hospitalized patients, (Centers for Disease Control and Prevention, Atlanta, GA, USA, 2022).

**Table 1 biomedicines-11-02055-t001:** SARS-CoV-2 Variants. Waves of COVID-19 infections driven by variants of SARS-CoV-2.

SARS-CoV-2 Variants	
	Alpha
	Beta
	Gamma
Origin (WA1/2020 strain) Delta	
	Omicron BA.1 BA.4 BQ.1 BA.2 BA.5 XBB

**Table 2 biomedicines-11-02055-t002:** Protective Efficacy of COVID-19 Vaccines Against Symptomatic Disease. Vaccine data against symptomatic disease in the United States during Phase 3 clinical trials before the emergence of the omicron variant.

Vaccine Dose Efficacy
Pfizer BNT162b2 (two injections) 95%
Moderna mRNA-1273 (two injections) 94%
Janssen Ad26.COV2.S (two injections) 94%
Janssen Ad26.COV2.S (one injection) 72%

## Data Availability

Not applicable.

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
