# Peer review of "COVID-19: Variants, Immunity, and Therapeutics for Non-Hospitalized Patients"

_biomedicines, 2023, doi:10.3390/biomedicines11072055_

Round 1

Reviewer 1 Report

Some comments for the authors to consider:

  1. The research topic, the scope of the study, and the main results could all be made clearer in the abstract. There is a lack of detail regarding the aims and procedures of the study in the present text.
  2. While the abstract does a good job of emphasizing how big the COVID-19 pandemic is, it fails to provide any specifics on the pandemic's impact on various populations or geographical areas. The abstract may be more helpful and instructive for readers if it included more background information.
  3. The introduction begins with some background on COVID-19 and its global influence. However, the study does not clearly articulate a research topic or objective that it seeks to achieve.
  4. Recent research addressing the current condition of COVID-19 variations and their impact is missing from the reference list, and the list as a whole is not comprehensive.
  5. Here, in Chapter 3, we take a look. There aren't any reputable sources to back up the claims made in this section. Particularly for numerical data and scientific evidence, citations would be beneficial.
  6. The section makes the immune response seem less complicated by going over the innate and adaptive immune responses in a simple way. The article might have provided more information about immune system components such as T cells, B cells, and antibodies.
  7. Included in Section 3 as well While the effectiveness of vaccines is touched on, no study on the effectiveness of the vaccine against COVID-19 or any of its variations is reviewed in any detail. The potential dangers and benefits of vaccines might also have been covered.
  8. There is room for improvement in making the most of Table 1's potential for clarity and aesthetic appeal. There are also sections of material that are too long and may be broken up to make the article easier to read.
  9. beginning with Section 5 Some of the claims made in this section are unsupported by citations. In lines 44–46, for instance, it is asserted that vaccination is the fundamental method for preventing the spread of the COVID-19 pandemic by eliciting T-cell immunity and neutralizing antibodies, but no sources are cited to back up this claim.
  10. On line 193, it is discussed that a new variety may emerge that will be completely resistant to the current vaccines and prevent serious sickness. Without supporting evidence, this assertion could provoke unnecessary alarm.
  11. In certain places, clarity is lacking, making it difficult to follow the argument. In line 225, for instance, the language "Pfizer's Paxlovid (Ritonavir/Nirmatrelvir) is an oral antiviral medicine granted Food and Drug Administration Emergency Use Authorization (FDA, EUA) in December 2021" should be rephrased to make it more understandable.
  12. Despite the progress that has been made, it is important to note the limits of the various treatments and vaccinations that have been discussed. For example, even though Pfizer's Paxlovid will be approved for use in December 2021, not all patients may be able to get it.
  13. Even though the article talks about the possible benefits of convalescent plasma therapy for COVID-19, it doesn't talk about the problems with getting plasma from donors, such as the risk of getting other diseases and the difficulty of getting enough plasma.
  14. But it doesn't say much about how this strategy could be used to treat COVID-19 or what the latest research shows about virus-mediated cellular senescence as a therapeutic target.
  15. It would be helpful to take a closer look at the studies mentioned in sections 8 and 9.This would mean talking about any biases or limits in the studies and putting the results into the bigger picture of COVID-19 therapy research.
  16. Line 356's phrasing "broader cross-sensitivity" is not easily understood and may cause confusion. And the line 357 sentence, "confer a longer term of protection," may be more precise if it defined "longer duration" more narrowly.
  17. At the end of the article, the need for novel treatments that can be tweaked to reduce the risk of hospitalizations and deaths caused by new coronavirus variations is mentioned. There is a lack of evidence and research to back up this assertion. Furthermore, the part is narrowly focused on the creation of new medicines and avoids discussing prevention, testing, and tracing connections between people who may be infected.
  18. I am not sure what the idea behind Figure 1 is. If it is meant to be a table, authors should follow journal formatting. Figure 2 quility is very low. 

  19. Tables don't follow the journal formatting styles 
  20. The figure legend should be comprehensive and clear without the need to read it from the text.
  21. the manuscript does not have a logical progression from beginning to end, which can make it hard for readers to comprehend the writers' points

Author Response

June 10, 2023

Editor-in-Chief

Biomedicines, MDPI

Dear Ladies and Gentlemen:

Attached is an updated and corrected manuscript, COVID-19: Variants, Immunity, and Therapeutics for Non-Hospitalized Patients by Cameron Y.S. Lee and Jon B. Suzuki. We greatly appreciate the time and effort provided by the editorial staff and reviewers to significantly improve the quality of this manuscript.

Corrections and updates in the body of the revised manuscript encompass virtually each comment of both reviewers.

  1. Abstract is now completely rewritten per suggestions by both reviewers. Topic defined, scope of study, and main points identified.
  2. Pandemic impact on selected populations groups identified and include both immunocompromised and elderly patients as higher risk.
  3. Research topic objective emphasized in abstract.
  4. Current COVID – 19 and variants identified and their impact.
  5. Chap 3, reference sources for claims in references # 27, 28, 29, 30, 31 and 32.
  6. More information on T Cells, plasma cells, B cells and Antibodies was added.
  7. Effective vaccines against COVID – 19 and variants added and show rather limited (4-6 months) of bioactivity.
  8. Table 1 improved for clarity and esthetics
  9. Section 5. Unsupported claims in references # 44-46.
  10. Line # 193 was eliminated, per reviewer # 1
  11. Line 225, Paxlovid… confusing information was clarified.
  12. Limits of COVID – 19 therapies and vaccines included as difficulty in access for some populations.
  13. Convalescent plasma therapies from donors are discussed in greater detail indicating challenges in identifying donors, preparation of serum, distribution.
  14. Strategies used to treat COVID-19 patients or research on virus-mediated cellular senescence as treatment presented.
  15. Sections 8 and 9, show some bias or limits in studies and put into the framework of the bigger picture of COVID-19 therapies.
  16. Line 356, Broader areas, and sensitivity was improved.
  17. Line 357, conferring longer terms of protection was changed to longer duration of protection.
  18. Figure 1 changed to a table. Figure 2 quality improved.
  19. Table need to follow formatting styles; Table 1 was eliminated and did not add to the manuscript.
  20. Figure legends were improved and more comprehensive. They are now able to be understood without referring to the text.
  21. Manuscript sections restyled to have more logical progression for clarification. Abstract was entirely rewritten.

Reviewer 2 Report

The review article by Cameron and Jon, “COVID-19: Variants, Immunity, and Therapeutics for Non-Hospitalized Patients” reports an update on the emerging variants of COVID-19 and the treatment options available to manage them. The authors advocate the development of new therapeutics, particularly vaccines with the ability for broader cross-sensitivity and longer-lasting COVID-19 protection. Overall, it is an insightful review and well-written.

Comments:

1.      The abstract does not adequately represent the content. The authors should make it more informative.

2.       Please provide a detailed explanation of table 1's contents in the write-up as the role of T cell activation and antibodies in different conditions (asymptomatic to death) may be unclear to readers.

Author Response

(The authors gave the same response as above.)

Round 2

Reviewer 1 Report

I would like to commend the authors for their diligent efforts in revising the manuscript. They have carefully considered each of the issues raised in the previous review and have made substantial improvements to the overall quality and clarity of the paper.

Author Response

/